# Clinical characteristics and risk factors for coronavirus disease 2019 (COVID-19) among patients under investigation in Thailand

**Jackrapong Bruminhent**[1], **Nattanon Ruangsubvilai**[2], **Jeff Nabhindhakara**[2], **Atiporn Ingsathit**[3,4], **Sasisopin Kiertiburanakul**[1]*

1 Division of Infectious Diseases, Department of Medicine, Faculty of Medicine Ramathibodi Hospital, Mahidol University, Bangkok, Thailand, 2 Faculty of Medicine Ramathibodi Hospital, Mahidol University, Bangkok, Thailand, 3 Division of Nephrology, Department of Medicine, Faculty of Medicine Ramathibodi Hospital, Mahidol University, Bangkok, Thailand, 4 Department of Clinical Epidemiology and Biostatistics, Faculty of Medicine Ramathibodi Hospital, Mahidol University, Bangkok, Thailand

* sasisopin.kie@mahidol.ac.th, sasisopin@hotmail.com

## Abstract

To manage coronavirus disease 2019 (COVID-19), a national health authority has implemented a case definition of patients under investigation (PUIs) to guide clinicians' diagnoses. We aimed to determine characteristics among all PUIs and those with and without COVID-19. We retrospectively reviewed clinical characteristics and risk factors for laboratory-confirmed COVID-19 cases among PUIs at a tertiary care center in Bangkok, Thailand, between March 23 and April 7, 2020. Reverse transcription-polymerase chain reaction for SARS-CoV-2 RNA was performed. There were 405 evaluable PUIs; 157 (38.8%) were men, with a mean age ± SD of 36.2 ± 12.6 years. The majority (68.9%) reported no comorbidities. There were 53 (13.1%) confirmed COVID-19 cases. The most common symptoms among those were cough (73.6%), fever (58.5%), sore throat (39.6%), and muscle pain (37.4%). Among these patients, diagnoses were upper respiratory tract infection (69.8%), viral syndrome (15.1%), pneumonia (11.3%), and asymptomatic infection (3.8%). Multivariate analysis identified close contact with an index case (OR, 3.49; 95%CI, 1.49–8.15; P = 0.004), visiting high-risk places (OR, 1.92; 95%CI, 1.03–3.56; P = 0.039), productive cough (OR, 2.03; 95%CI, 1.05–3.92; P = 0.034), and no medical coverage (OR, 3.91; 95%CI, 1.35–11.32; P = 0.012) as independent risk factors for COVID-19 among the PUIs. The majority had favorable outcomes, though one (1.9%) died from severe pneumonia. COVID-19 was identified in 13% of PUIs defined per a national health authority's case definition. History of contact with a COVID-19 patient, visiting a high-risk place, having no medical coverage, and productive cough may identify individuals at risk of COVID-19 in Thailand.

## Introduction

Coronavirus disease 2019 (COVID-19) is an emerging respiratory tract infection caused by severe acute respiratory syndrome coronavirus-2 (SARS-CoV-2), a novel coronavirus initially

**Data Availability Statement:** All relevant data are within the manuscript and its Supporting Information files.

**Funding:** The author(s) received no specific funding for this work.

**Competing interests:** No contributors have any conflicts of interest.

reported in China and later spreading worldwide [1]. In January 2020, the World Health Organization (WHO) declared COVID-19 a Public Health Emergency of International Concern [2]. COVID-19 patients can present as asymptomatic, mild upper respiratory tract disease or potentially severe pneumonia. Consequentially, those with severe infection are at potential risk for acute hypoxemic respiratory failure requiring mechanical ventilation, a condition with substantially high morbidity and mortality. The approximate mortality rate has ranged from 1% to 10% depending on patients' clinical presentations and the allocation of medical resources, varying among resource-adequate and constrained settings [3, 4]. On January 12, 2020, the Ministry of Public Health (MoPH), Thailand, reported the first imported patient who tested positive for COVID-19 outside China [5]. As of April 25, 2020, a total of 2,907 people in Thailand were diagnosed with COVID-19, with a mortality rate of approximately 1.8% [6]. Among the cases, the greatest risk factors of contracting COVID-19 were found to be close contact with an index case or a history of travel to a high-risk area. Based on a number of recent case reports, these seem to be crucial clues for a diagnosis of COVID-19 in Thailand [7, 8]. The MoPH has therefore set out a definition of patients under investigation (PUIs) to identify patients at risk of contracting COVID-19; this is based on the United States Centers for Disease Control and Prevention (CDC) guidelines. A stratified investigation was attempted to better identify patients at risk and who needed investigation based on capacity and accessibility to nucleic acid amplification testing, which was considered unaffordable for some areas of Thailand. Specific risk factors could assist clinicians in predicting which PUIs were infected with COVID-19. Although there was a previous case series of COVID-19 patients who were hospitalized [9], there has been no study focused on this population. Moreover, the rate of cases testing positive for COVID-19 based on the case definition had not been assessed and clinical characteristics and risk factors for non-infected versus infected PUIs has not been explained. We therefore aimed at a large-scale investigation of this all entities and expected to define better criteria for identifying cases. This would lead to improved diagnoses, prompt therapy, and infection control.

## Material and methods

The present study was conducted at Ramathibodi Hospital, a 1,200-bed, university hospital in the center of the Bangkok Metropolitan Area in Thailand. We conducted a retrospective review examining for COVID-19 in PUIs aged ≥15 years covering March 23 to April 7, 2020, when the highest rate of cases was reported in Thailand. A list of PUIs was retrieved from a database of the Infection Prevention and Control Services at our hospital.

### Definitions of PUIs

The Department of Disease Control, MoPH, on March 2, 2020, defined PUIs as follows [10]. First, these patients have a history of fever or fever ≥37.5˚C (99.5˚F) and one respiratory tract symptom (e.g., cough, runny nose, sore throat, tachypnea, dyspnea, difficulty breathing), and during the 14 days before developing symptoms they: (a) traveled to or from Thailand or lived in an area with a report of an ongoing outbreak of COVID-19; (b) worked and had close contact with tourists, worked in a crowded place, or had contact with many people; (c) had contact with confirmed patients or with respiratory droplets of patients suspected of or confirmed with having COVID-19, and without appropriate protective equipment; or (d) had a history of going to a community location or a place with groups of people (e.g., market, department store, hospital) as announced by the Provincial Communicable Disease Committee. Second, they are pneumonia patients with a history of one of the following: (a) had close contact with a COVID-19 patient; (b) had unexplained pneumonia and the clinical condition did not

improve within 48–72 hours; or (c) had pneumonia with a profile consistent with that of COVID-19. Third, the patients are medical personnel with a history of fever or fever ≥37.5°C (99.5°F) and one respiratory tract symptom, and the physician in charge or the communicable disease control officer advised an examination for SARS-CoV-2. Last, there was detection of a group of cases in the same place, within the same week, and with an epidemiologic connection.

We also included those who were not entirely matching the case definition but were deemed to be at risk based on their substantial exposure or typical symptoms (or both). The latter was determined by a certified infectious diseases specialist designated to decide who should be investigated as a PUI.

Per practical flow, each PUI was initially determined as a non-severe or severe case based on our institution criteria developed by members of the Infectious Diseases Division in conjunction with the Pulmonary and Critical Care Division. Non-severe PUIs underwent an interview and physical examination at a designated acute respiratory infection clinic at the Faculty of Medicine Ramathibodi Hospital. Severe PUIs were defined as one of the following: peripheral capillary oxygen saturation (SpO$_2$) <92% with room air, SpO$_2$ 92%–95% (room air) with respiratory rate (RR) >30 breaths/min, or RR <30 breaths/min with signs of impending respiratory failure. Those classified as severe were directly admitted to an airborne infection isolation room (AIIR) in an intensive care unit. All confirmed COVID-19 patients were mandatorily admitted to the hospital.

Demographic data including sex, age, home address, occupation, health insurance scheme enrollment, underlying disease, and presenting symptoms were obtained by reviewing medical records. Those with risk of SARS-CoV-2 infection in accordance with the definitions of PUIs as per the MoPH (as described above) were also retrieved and reviewed. We also collected complete blood count, chemistry laboratory testing results, chest X-ray findings, and the results of SARS-CoV-2 RNA detection. We divided patients' final diagnoses into two groups—patients with positive and negative results of SARS-CoV-2 RNA—to determine clinical characteristics and risk factors for COVID-19.

Members of the Infectious Diseases Division developed the treatment regimen at our institution in conjunction with the Pulmonary and Critical Care Division; this was guided by the Department of Disease Control, MoPH [11]. (Hydroxy)chloroquine and boosted protease inhibitors, either boosted lopinavir or darunavir, were given to those with mild (with comorbidities) and moderate symptoms. Favipiravir, an anti-viral agent, was added for those diagnosed with pneumonia. Supportive treatment was offered for all patients.

## SARS-CoV-2 reverse transcription polymerase chain reaction (PCR)

Nasopharyngeal and throat swabs or endotracheal aspirates from those who were intubated were collected from PUIs using COPAN FLOQSwabs, and a sterile tube containing COPAN's Universal Transport Medium (COPAN Diagnostics Inc.). Viral RNA was extracted from the samples using MagDEA Dx reagents (Precision System Science Co., Ltd.) with a fully nucleic acid extraction system. The detection of SARS-CoV-2 was performed using reverse transcription-PCR (RT-PCR); this was performed using a CFX96 Real-Time PCR Detection system (Bio-Rad Laboratories, Inc.). Amplification of SARS-CoV-2 *ORF1ab* and *N* gene fragments, using a SARS-CoV-2 Nucleic Acid Diagnostic Kit (Sansure Biotech Inc.), was approved by the National Medical Products Administration and certified by the China Food and Drug Administration [12]. Physicians were allowed to investigate for other respiratory viruses, such as influenza or other pathogens, for patients deemed to be at risk.

The study protocol was approved by the Institutional Review Board (IRB) of the Faculty of Medicine Ramathibodi Hospital, Mahidol University, Bangkok, Thailand, with the provisions

of the Good Clinical Practice Guidelines and the Declaration of Helsinki (approval number: COA. MURA2020/557). All data were fully anonymized before accessed and the IRB waived the requirement for informed consent.

## Statistical analysis

Median values (with interquartile range, IQR) or mean (with standard deviation, SD) were used to describe the patients' baseline characteristics, and laboratory investigations for continuous data and frequency were used for categorical data. A chi-square test or Fisher's exact test and Wilcoxon rank-sum test were used to compare categorical variables and continuous variables between the two groups, respectively. Univariate and multivariate logistic regression analyses were performed to determine the factors associated with positive results for SARS-CoV-2 RNA. Variables that presented $P < 0.05$ in the univariate analysis were considered in a multivariate logistic regression model after assessment of multicollinearity of variance inflation factors. Variables were selected into a multiple logistic regression model with forward stepwise selection, and those that attained significance were retained in the model. The odds ratio (OR) and its 95% confidence interval (CI) were estimated. $P < 0.05$ was considered statistically significant. All statistical analyses were performed using Stata statistical software version 15.1 (StataCorp. 2017. Stata Statistical Software: Release 15. College Station, TX: StataCorp LLC.).

## Results

### PUIs

A total of 414 patients during the study period were investigated for COVID-19. Table 1 shows the baseline characteristics of all PUIs. Nine were excluded because of incomplete data (n = 5) and because PCR for SARS-CoV-2 was not tested (n = 4). Among the remaining 405 evaluable patients, 157 (38.7%) were men, with a mean age ± SD of 36.2 ± 12.6 years. The majority (96.8%) were of Thai ethnicity and (85.2%) lived in the Bangkok Metropolitan Area. A total of 149 (36.8%) patients were unemployed and 297 (73.3%) had to self-pay their full medical expenses. Only around one-third (31.1%) had underlying diseases, including allergic rhinitis (6.4%), diabetes mellitus (5.7%), hypertension (4.2%), and dyslipidemia (1.7%). Few patients (2.5%) had immunocompromised conditions. There were 347 (85.7%) and 58 (14.3%) patients who were fulfilled the criteria and designated as a PUI, respectively. Twenty-six (6.4%) severe PUIs and 379 (93.6%) non-severe PUIs were classified as the aforementioned criteria.

### Diagnosed COVID-19 patients

Among 400 (98.8%) patients underwent nasopharyngeal and throat swabs and 5 (1.2%) patients provided endotracheal aspirates for SARS-CoV-2 PCR, a total of 53 (13%) patients were confirmed as having COVID-19; 18 (34%) patients were men with mean age ± SD of 36.3 ± 10.2 years. The majority were of Thai ethnicity (98.1%), lived in the Bangkok Metropolitan Area (84.9%), were unemployed (39.6%), and worked at a restaurant (34%). Most (92.4%) did not have medical coverage. Most (81.1%) had no underlying diseases and none were immunocompromised. Confirmed COVID-19 patients were classified as having upper respiratory tract infection (69.8%), viral syndrome (15.1%), pneumonia (11.3%), or asymptomatic infection (3.8%). The patients were admitted into an AIIR located in the general ward (98.1%) or intensive care unit (ICU) (1.9%). Among the latter, 5 (1.2%) patients underwent endotracheally intubation on arrival due to acute respiratory failure and therefore endotracheal aspirates were collected accordingly.

**Table 1. Baseline characteristics of 405 patients under investigation.**

| Variable | Value |
|---|---|
| Male, n (%) | 157 (38.8) |
| Mean (standard deviation) age, years | 36.2 (12.6) |
| Nationality, n (%) | |
| Thai | 392 (96.8) |
| Non-Thai | 13 (3.2) |
| Home region in Thailand, n (%) | |
| Bangkok Metropolitan Area | 345 (85.2) |
| Central | 17 (4.2) |
| North | 6 (1.5) |
| East | 2 (0.5) |
| Northeast | 23 (7.4) |
| South | 12 (2.9) |
| Occupation, n (%) | |
| Unemployed | 149 (36.8) |
| Healthcare worker | 28 (6.9) |
| Merchant | 17 (4.2) |
| Waitstaff or bar worker | 63 (15.6) |
| Public transportation worker | 32 (7.9) |
| Freelance | 40 (9.9) |
| Office worker | 37 (9.1) |
| Student | 39 (9.6) |
| Medical coverage | 108 (26.7) |
| Underlying diseases, n (%) | 126 (31.3) |
| Diabetes mellitus | 23 (5.7) |
| Hypertension | 17 (4.2) |
| Dyslipidemia | 7(1.7) |
| Cardiovascular disease | 2 (0.5) |
| Cancer | 4 (1.0) |
| Chronic liver disease | 2 (0.5) |
| Asthma | 8 (1.9) |
| Allergic rhinitis | 26 (6.4) |
| Others | 37 (9.1) |

Compared with non-COVID-19 patients (Table 2), there were no differences in the proportion of male sex, median age, nationality, home address, or underlying disease between groups (P >0.05 for all). Those with COVID-19 lacked medical coverage at a significant rate compared with those without COVID-19 (92.4% vs. 7.04%; P = 0.001). Compared with non-COVID-19, those with COVID-19 reported a history of close contact with an index case (86.8% vs. 63.1%; P = 0.001), visiting a crowded place, or attending an activity where people gathered, in the 14 days before symptom onset (41.5% vs. 26.7%; P = 0.026). However, there was no significant difference regarding history of traveling abroad in the 14 days before symptom onset (1.9% vs. 8.2%; P = 0.100).

Both groups reported having symptoms on arrival at a rate as high as 94%. Those with COVID-19 complained significantly more frequently of productive cough (34% vs. 21.6%; P = 0.047). They also experienced slightly more subjective fever (58.5% vs. 44.3%; P = 0.054) and anosmia (5.7% vs. 1.1%; P = 0.051), though slightly less diarrhea (3.8% vs. 12.5%; P = 0.062). Other respiratory symptoms showed no significant difference, including dry

**Table 2. Baseline characteristics of patients under investigation with and without COVID-19.**

| Variable | COVID-19 (n = 53) | Non-COVID-19 (n = 352) | P-value |
|---|---|---|---|
| Mean (standard deviation) age, years | 36 (10) | 36 (12) | 0.417 |
| Male, n (%) | 18 (34.0) | 139 (39.5) | 0.441 |
| Thai nationality, n (%) | 52 (98.1) | 340 (96.6) | 1.000 |
| Lived in Bangkok Metropolitan Area, n (%) | 45 (84.9) | 300 (85.2) | 0.951 |
| Unemployed, n (%) | 21 (39.6) | 128 (36.4) | 0.646 |
| Medical coverage, n (%) | 4 (7.6) | 104 (29.6) | 0.001 |
| Traveled abroad in the 14 days before symptom onset, n (%) | 1 (1.9) | 29 (8.2) | 0.155 |
| Contact, in the 14 days before symptom onset, with a person who traveled abroad, n (%) | 3 (5.7) | 26 (7.4) | 1.000 |
| Contact, in the 14 days before symptom onset, with a person who traveled from another province, n (%) | 0 (0.0) | 1 (0.3) | 1.000 |
| Visit to high-risk place, n (%) | 22 (41.5) | 94 (26.7) | 0.026 |
| Contact with a COVID-19 patient, n (%) | 46 (86.8) | 222 (63.1) | 0.001 |
| Contact with person who was diagnosed with pneumonia with unknown cause, n (%) | 0 (0.0) | 1 (0.3) | 1.000 |
| Symptomatic, n (%) | 50 (94.3) | 330 (93.8) | 1.000 |
| Median (IQR) duration of symptoms, days | 4 (2–7) | 3 (2–7) | 0.288 |
| Clinical manifestations, n (%) | | | |
| Fever | 31 (58.5) | 156 (44.3) | 0.054 |
| Dry cough | 21 (39.6) | 179 (50.8) | 0.127 |
| Productive cough | 18 (34.0) | 76 (21.6) | 0.047 |
| Nasal congestion | 2 (3.8) | 6 (1.7) | 0.282 |
| Rhinorrhea | 12 (22.6) | 117 (33.2) | 0.123 |
| Sore throat | 21 (39.6) | 164 (46.6) | 0.342 |
| Shortness of breath | 15 (28.3) | 111 (31.5) | 0.636 |
| Myalgia | 20 (37.7) | 100 (28.4) | 0.166 |
| Nausea/vomiting | 2 (3.8) | 20 (5.7) | 0.753 |
| Headache | 11 (20.8) | 62 (17.6) | 0.579 |
| Fatigue | 2 (3.8) | 41 (11.6) | 0.083 |
| Diarrhea | 2 (3.8) | 44 (12.5) | 0.062 |
| Anosmia | 3 (5.7) | 4 (1.1) | 0.051 |
| Vital signs | | | |
| Median (IQR) temperature, degrees Celsius | 37 (1) | 37 (0) | 0.944 |
| Median (IQR) pulse, beats per minute | 78 (8.5) | 98 (24) | 0.008 |
| Median (IQR) respiratory rate, breaths per minute | 23 (10) | 22 (4) | 0.948 |
| Median (IQR) systolic blood pressure, mmHg | 110 (10) | 133 (44) | 0.008 |
| Median (IQR) diastolic blood pressure, mmHg | 64 (10) | 80 (13) | 0.013 |
| Median (IQR) SpO$_2$, % | 96.5 (4.5) | 98 (2) | 0.042 |
| Abnormal physical examination, n (%) | 8 (15.1) | 75 (21.3) | 0.296 |
| Abnormal HEENT examination, n (%) | 7 (13.2) | 62 (17.6) | 0.426 |
| Injected pharynx, n (%) | 6 (11.3) | 57 (16.2) | 0.362 |
| Enlarged tonsils, n (%) | 2 (3.8) | 23 (6.5) | 0.758 |
| Skin rash, n (%) | 0 (0.0) | 2 (0.6) | 1.000 |
| Abnormal lung examination, n (%) | 2 (3.8) | 15 (4.3) | 1.000 |
| Underlying disease, n (%) | 10 (18.9) | 116 (33) | 0.039 |
| Immunocompromised condition, n (%) | 0 (0) | 10 (2.8%) | 0.372 |
| Chest X-ray findings, n (%) | | | 0.298 |
| Normal | 1 (14.3) | 16 (45.7) | |
| Patchy opacity | 2 (28.6) | 5 (14.3) | |
| Reticular/interstitial opacity | 4 (57.1) | 9 (25.7) | |

*(Continued)*

**Table 2.** (Continued)

| Variable | COVID-19 (n = 53) | Non-COVID-19 (n = 352) | P-value |
|---|---|---|---|
| Old infiltration | 0 (0.0) | 4 (11.4) | |
| Others | 0 (0.0) | 1 (2.9) | |

SpO$_2$: peripheral capillary oxygen saturation; HEENT: head, ear, eye, nose, and throat; IQR: interquartile range

cough, nasal congestion, rhinorrhea, sore throat, myalgia, shortness of breath, or gastrointestinal symptoms, and headache (P >0.05, all). Two patients presented with skin rashes; both were in the non-COVID-19 group. Regarding vital signs, those with COVID-19 had significantly lower median pulse (78 beats/min vs. 98 beats/min; P = 0.008), systolic blood pressure (110 mmHg vs. 133 mmHg; P = 0.008), diastolic blood pressure (64 mmHg vs. 80 mmHg; P = 0.013), and oxygen saturation (96.5% vs. 98%; P = 0.042). Body mass index did not significantly differ (P >0.05). The median duration of presenting symptoms did not significantly differ (4 days vs. 3 days; P = 0.288). Only 15% and 21% of COVID-19 and non-COVID-19 patients, respectively, had an abnormal physical exam. An abnormal pharyngeal exam, including injected pharynx and enlarged tonsils, were reported in 13.2% and 17.6%, respectively (P = 0.426). Abnormal chest examination was seen in 3.8% and 4.3% (P = 1.000), respectively.

From the standpoint of laboratory testing results (Table 3), those with COVID-19 presented with significantly lower median white blood cell counts, at 6,100 cells/mm$^3$, compared with 9,600 cells/mm$^3$ in non-COVID-19 patients (P = 0.002). The median percentages of lymphocytes (25.5% vs. 14%; P = 0.056) and monocytes (10.5% vs. 7%; P = 0.030) were slightly more prominent in COVID-19 patients. Prevalence of lymphopenia and thrombocytopenia did not significantly differ. Levels of liver function tests, lactate dehydrogenase, C-reactive protein, and procalcitonin were indistinguishable between the groups (P >0.05 for all). Abnormal chest X-ray was more frequently observed in the COVID-19 group, including patchy opacity

**Table 3. Laboratory results of patients under investigation with and without COVID-19.**

| Variable, median (IQR) | COVID-19 (n = 6) | Non-COVID-19 (n = 23) | P-value |
|---|---|---|---|
| White blood cells, cells/mm$^3$ | 6,100 (4,500–6,800) | 9,600 (7,500–12,900) | 0.002 |
| Neutrophil, % | 62 (43–74) | 76 (68–87) | 0.101 |
| Lymphocyte, % | 26 (20–44) | 14 (6–23) | 0.056 |
| Monocyte, % | 11 (9–11) | 7 (4–8) | 0.030 |
| Hemoglobin, g/dL | 13 (13–13) | 11.5 (11–14) | 0.372 |
| Platelet, cells/mm$^3$ | 236,000 (180,000–274,000) | 260,000 (177,000–306,000) | 0.484 |
| Blood urea nitrogen, mg/dL | 7 (7–9) | 12 (9–16) | 0.172 |
| Creatinine, mg/dL | 1 (1–1) | 1 (1–1) | 0.089 |
| Aspartate aminotransferase, U/L | 44 (42–50) | 33 (22–60) | 0.650 |
| Alanine aminotransferase, U/L | 23 (23–26) | 28.5 (20–82) | 0.626 |
| Total bilirubin, mg/dL | 0.4 (0.3–0.6) | 0.8(0.5–0.9) | 0.125 |
| Direct bilirubin, mg/dL | 0.2 (0.2–0.3) | 0.3 (0.2–0.6) | 0.173 |
| Alkaline phosphatase, U/L | 53 (39–98) | 78 (71–109) | 0.174 |
| Gamma-glutamyl transferase, U/L | 56 (31–88) | 46 (38–144) | 0.715 |
| Albumin, g/L | 31.5 (29.5–35) | 34 (26–39) | 0.571 |
| Lactate dehydrogenase, U/L | 222 (136–308) | 351 (199–490) | 0.317 |
| C-reactive protein, mg/L | 67 (5–129) | 88 (7–95) | 1.000 |

IQR: interquartile range

(28%) and reticular/interstitial opacity (57%). Those with COVID-19 were more likely to be diagnosed with pneumonia compared with upper respiratory tract infection (11.3% vs 3.1%, p = 0.029). Among 352 (86.9%) PUIs who did not have a positive result for COVID-19, 40 (11.4%) patients underwent further investigations. There were seven patients in the non-COVID-19 group diagnosed with infections with non-COVID pathogens: influenza virus (n = 2), *Pneumocystis jirovecii* (n = 1), *Haemophilus influenzae* (n = 2), *Klebsiella pneumoniae* (n = 1), and *Staphylococcus aureus* (n = 1). The majority of diagnosed COVID-19 patients had favorable outcomes, though one (1.9%) died from severe pneumonia.

### Risk factors for COVID-19

Univariate analysis (Tables 4 and 5) showed visiting high-risk places (OR, 1.94; 95%CI, 1.07–3.53; P = 0.028), close contact with a COVID-19 patient (OR, 3.85; 95%CI, 1.69–8.77; P = 0.001), productive cough (OR, 1.88; 95%CI, 1.00–3.48; P = 0.049), and anosmia (OR, 5.22; 95%CI, 1.13–24.01; P = 0.034) were significantly more likely to be present in those with COVID-19. Those who had medical coverage (OR, 0.19; 95%CI, 0.07–0.55; P = 0.002) and underlying disease (OR, 0.47; 95%CI, 0.23–0.98; P = 0.043), more rapid pulse (OR, 0.95; 95%CI, 0.90–0.99; P = 0.028), higher oxygen saturation (OR, 0.79; 95%CI, 0.65–0.96; P = 0.017), higher diastolic blood pressure (OR, 0.83; 95%CI, 0.70–0.98; P = 0.030), and higher white blood cell count per 100 cells/mm$^3$ (OR, 0.95; 95%CI, 0.90–1.00; P = 0.038) were significantly less likely to have COVID-19.

Multivariate analysis (Table 6) found contact with an index case (OR, 3.49; 95%CI, 1.49–8.15; P = 0.004), visiting high-risk places (OR, 1.92; 95%CI, 1.03–3.56; P = 0.039), productive cough (OR, 2.03; 95%CI, 1.05–3.92; P = 0.034), and no medical coverage (OR, 3.91; 95%CI, 1.35–11.32; P = 0.012) were independently associated with COVID-19.

### Discussion

The present study appears to be one of the first and largest to evaluate incidence of and predictors for COVID-19 in a setting in which patients were investigated under the impetus of a national health authority's criteria. We found approximately one in eight PUIs had COVID-19 confirmed by molecular testing. Most patients had no comorbidities, presented with upper respiratory tract infection, and had a favorable outcome. Apart from close contact with an infected case and visiting high-risk places, we found that having no medical coverage and presenting with productive cough were predictors of being diagnosed with COVID-19 among PUIs.

SARS-CoV-2 is an emerging respiratory virus that commonly causes no or mild respiratory tract infection and is occasionally complicated by severe pneumonia [1]. The strategy for identifying index case has varied among different settings depending on risk and exposure. A targeted approach rather than universal testing may be more practical in areas where resources are limited. In Thailand, case definition-driven cases were previously used for diagnosis of Middle East respiratory syndrome coronavirus (MERS-CoV); however, the spread and impact of MERS-CoV were considered less than with SARS-CoV-2 [13]. We were able to use our national authority's case definition to identify COVID-19 cases. Additionally, we validated histories of close contact with an index case and of visiting high-risk places, which were already included in the criteria. Chen et al. reported that half of patients with a history of exposure to the seafood market suspected to be the sources of the virus, and close contact with a COVID-19-infected individual, were major risk factors among people who lived in Wuhan, Hubei, China during the initial outbreak [14]. In Thailand, people who had been to bars and boxing events were the first two clusters of cases reported in the heart of the Bangkok Metropolitan

**Table 4. Univariate logistic regression analysis of associated factors for COVID-19 (clinical characteristics and physical examinations).**

| Variable | Odds ratio | 95% confidence interval | P-value |
|---|---|---|---|
| Male | 0.79 | 0.43–1.45 | 0.442 |
| Age | 1.00 | 0.98–1.02 | 0.938 |
| Thai ethnicity | 0.54 | 0.70–4.28 | 0.564 |
| Bangkok Metropolitan Area resident | 1.08 | 0.88–1.33 | 0.446 |
| Employed | 0.87 | 0.48–1.57 | 0.647 |
| Medical coverage | 0.19 | 0.07–0.55 | 0.002 |
| Underlying disease | 0.47 | 0.23–0.98 | 0.043 |
| Travel abroad in the 14 days before symptom onset | 0.21 | 0.03–1.61 | 0.134 |
| Contact, in the 14 days before symptom onset, with a person who traveled abroad | 0.75 | 0.22–2.58 | 0.651 |
| Going to high-risk places | 1.94 | 1.07–3.53 | 0.028 |
| Contact with a COVID-19 patient | 3.85 | 1.69–8.77 | 0.001 |
| Symptomatic | 1.11 | 0.32–3.85 | 0.868 |
| Duration of symptom | 0.99 | 0.92–1.07 | 0.897 |
| Clinical manifestations | | | |
| Fever | 1.77 | 0.98–3.18 | 0.056 |
| Dry cough | 0.63 | 0.35–1.14 | 0.130 |
| Productive cough | 1.87 | 1.00–3.48 | 0.049 |
| Nasal congestion | 2.26 | 0.44–11.51 | 0.326 |
| Rhinorrhea | 0.59 | 0.30–1.16 | 0.126 |
| Sore throat | 0.75 | 0.42–1.36 | 0.343 |
| Shortness of breath | 0.86 | 0.45–1.62 | 0.636 |
| Myalgia | 1.53 | 0.84–2.79 | 0.168 |
| Nausea/vomiting | 0.65 | 0.15–2.87 | 0.571 |
| Headache | 1.23 | 0.60–2.51 | 0.580 |
| Fatigue | 0.30 | 0.07–1.27 | 0.101 |
| Diarrhea | 0.27 | 0.06–1.17 | 0.080 |
| Anosmia | 5.22 | 1.13–24.01 | 0.034 |
| Vital signs | | | |
| Temperature | 1.06 | 0.59–1.89 | 0.851 |
| Pulse | 0.94 | 0.90–0.99 | 0.028 |
| Respiratory rate | 1.01 | 0.86–1.19 | 0.883 |
| Systolic blood pressure | 0.88 | 0.76–1.02 | 0.088 |
| Diastolic blood pressure | 0.83 | 0.70–0.98 | 0.030 |
| $SpO_2$ | 0.79 | 0.65–0.96 | 0.017 |
| Abnormal physical examination | 0.66 | 0.30–1.45 | 0.299 |
| Abnormal HEENT examination | 0.71 | 0.31–1.65 | 0.428 |
| Injected pharynx | 0.66 | 0.27–1.62 | 0.364 |
| Tonsil enlargement | 0.56 | 0.13–2.45 | 0.442 |
| Abnormal lung examination | 0.88 | 0.20–3.97 | 0.869 |

$SpO_2$: peripheral capillary oxygen saturation; HEENT: head, ear, eye, nose, and throat

Area in early March 2020 [15]. We also discovered that having no medical coverage and having productive cough, which were not included in the case definition, were novel risk factors. Lack of medical coverage in Thailand would likely reflect challenging socioeconomic status,

**Table 5. Univariate logistic regression analysis of associated factors for COVID-19 (laboratory results).**

| Variables | Odds ratio | 95% confidence interval | P-value |
|---|---|---|---|
| White blood cells (per 100 cells/mm$^3$) | 0.95 | 0.90–1.00 | 0.038 |
| Percentage of neutrophils | 0.95 | 0.90–1.01 | 0.099 |
| Percentage of lymphocytes | 1.07 | 1.00–1.16 | 0.056 |
| Percentage of monocytes | 1.14 | 0.92–1.42 | 0.222 |
| Hemoglobin | 1.44 | 0.77–2.68 | 0.257 |
| Platelet <150,000/mm$^3$ | 2.24 | 0.23–21.91 | 0.489 |
| Blood urea nitrogen | 0.95 | 0.82–1.10 | 0.492 |
| Aspartate aminotransferase | 1.00 | 0.96–1.04 | 0.999 |
| Alanine aminotransferase | 0.97 | 0.91–1.03 | 0.321 |
| Total bilirubin | 0.02 | 0.00–7.53 | 0.198 |
| Alkaline phosphatase | 0.97 | 0.93–1.02 | 0.284 |
| Albumin | 0.98 | 0.83–1.15 | 0.777 |
| Lactate dehydrogenase | 0.99 | 0.98–1.01 | 0.318 |
| C-reactive protein | 1.00 | 0.97–1.04 | 0.936 |
| Abnormal chest X-ray | 1.15 | 0.62–2.13 | 0.665 |

such as living in crowded households and certain types of workplaces. Our study supported possible human-to-human transmission in a closed environment and among family members [7, 16, 17]. A national policy is also implemented to monitor those who were closely contacted with an index patient (COVID-19 patient) and complimentarily investigated for COVID-19 should new symptoms occur. Individuals who pay for their own medical care out-of-pocket may also have superior access to medical services. Furthermore, productive cough remained associated with COVID-19 after adjusting for other covariables. Sputum production was previously reported in roughly one-third of a cohort roughly one-third apart from dry cough [18]. An unexpected caveat is although the male gender predominance was observed among several cohorts regarding vulnerability to COVID-19, our result instead revealed female gender is more frequently diagnosed. A reason to explain this disparity has been proposed but not entirely clear, however, an outcome seemed indifferent [19, 20]. Furthermore, a greater proportion of female PUIs in our cohort likely from a coincidence or possibly more attention in their health conditions was more prominent among the female population.

Among confirmed COVID-19 patients, our patient data regarding clinical symptoms and signs were both comparable and somewhat different from those in previous reports from China [14, 21]. Measured body temperatures were similar to those in previous studies that were found more likely to be observed at admission [18]. Another study found subjective fever was more likely to be reported among COVID-19 patients [21]. Respiratory symptoms were indistinguishable between those with and without COVID-19 in our cohort, except for productive cough, which was contrary to the previous cohort. Patients diagnosed with COVID-19 in China reported having cough in up to 80% of cases; and more specifically, dry cough in

**Table 6. Multivariate regression analysis of risk factors for COVID-19.**

| Variables | Odds ratio | 95% confidence interval | P-value |
|---|---|---|---|
| No medical coverage | 3.91 | 1.35–11.32 | 0.012 |
| Visiting high-risk place | 1.92 | 1.03–3.56 | 0.039 |
| Close contact with COVID-19 patient | 3.49 | 1.49–8.15 | 0.004 |
| Expectoration | 2.03 | 1.05–3.92 | 0.034 |

nearly 60% [14, 21]. We also found anosmia was more likely to be reported among confirmed COVID-19 cases; it was significantly more frequent among COVID-19 patients than among influenza patients in a pilot case-control study [22]. To the contrary, gastrointestinal symptoms such as diarrhea, nausea, and vomiting were more likely to present in non-COVID-19 patients. Chen et al reported gastrointestinal symptoms in only 1%–2% of COVID-19 pneumonia patients [14]. A case series of Thai COVID-19 hospitalized patients showed fever and respiratory symptoms were not present at significant levels even when all patients were diagnosed with radiologically confirmed pneumonia [9]. This agreed with our finding that respiratory symptoms were not prominent. Wang et al. reported anosmia, dyspnea, sore throat, dizziness, and abdominal pain were more common among COVID-19 patients who were admitted to an intensive care unit (ICU) in China [21]. A case definition focused on fever and respiratory signs and symptoms may inadequately detect some cases, especially those later in the disease course. We found faster heart rate, lower oxygen saturation, and borderline low blood pressure were associated with COVID-19. There was no such association, however, in a logistic analysis. This was likely owing to the small number of patients.

Our study confirmed that most of the infected patients were asymptomatic or had mild symptoms. They therefore could pose a greater risk of transmitting the disease within a community. Asymptomatic patients with upper respiratory specimens testing positive for SARS-CoV-2 transmission were previously reported [23, 24]. Co-infection with other pathogens coinciding with SARS-CoV-2 is plausible; however, the patients in the present study did not receive further testing after COVID-19 because they all had mild symptoms and further workups did not show evident impact on disease management [9].

Regarding laboratory investigations, COVID-19 patients presented with significantly lower WBC count compared with non-COVID-19 patients. Lymphocytosis was commonly seen in viral infections in general. However, we observed the percentages of lymphocytes and monocytes were slightly more prominent in the COVID-19 patients. Our data did not reveal lymphopenia or thrombocytopenia because the patients in our study had slightly less severe cases than those in another cohort [18]. Lymphocyte count was lower in ICU compared with non-ICU COVID-19 patients [21]. Among Thai COVID-19 patients who were hospitalized, those with severe COVID-19 tended to have leukopenia, lymphopenia, and slight thrombocytopenia [9]. Other tests, including liver function, lactate dehydrogenase, inflammatory markers such as C-reactive protein (CRP), and procalcitonin, were unable to distinguish COVID-19 and non-COVID-19 cases in our PUI cohort.

The nucleic acid amplification testing used in our hospital detects SARS-CoV-2 by targeting the specific conserved sequence approved by the WHO [12, 25]. The primers and probes target the *ORF1ab* and *N* genes of SARS-CoV-2. COVID-19 serological tests were not performed because the test was unavailable during the study period. Inability to access medical resources and high costs of PCR testing are potential barriers to implementing universal screening in a general population.

This study had several limitations. First, recall bias was inevitable based on the nature of the retrospective study. A record-keeping form implemented at acute respiratory infection clinics would better assist physicians in gathering important information, performing a more thorough review, and collecting as much data as possible. Second, risk factors identified in our cohort are more specific for the local population investigated and may not generalizable. We do, however, encourage clinicians to investigate those variables for their relevant populations because lifestyle and exposure may vary among regions. Third, some COVID-19 patients may have been overlooked, especially those presenting with atypical or non-respiratory symptoms as have been reported in the literature [26, 27]. Fourth, our study was covered for only 2-week period due to an epidemiological characteristic of COVID-19 in Thailand is relatively brief.

Therefore, we tentatively selected this specific period when the majority of PUIs attended healthcare facilities for an investigation when the information regarding some specific risk factors was completely collectable and deem interpretable. Finally, diagnostic criteria and identification of PUIs have evolved as more data have emerged. The case definition should be revised based on the current situation in Thailand. A previous version of the definition was more specific for imported cases while <7% of PUIs were detected at airport screening [15]. We also acknowledge incomplete data gathering and examination among those with mild cases, due to an attributable effect of the contagious disease. However, the present study is one of the first conducted in Thailand involving an intervention on PUIs using a screening test to include those at risk of SARS-CoV-2 infection acquisition. Our hospital also, by far, conducts the most virology studies of any institution in Thailand. We believe the aforementioned variables could better stratify those at risk in addition to a criterion. More importantly, we encourage a case definition combined with thorough history-taking to improve sensitivity of this definition.

## Conclusions

A national health authority criterion could potentially predict COVID-19 diagnosis during an outbreak setting in Thailand. Clinicians should be aware of those who have no medical care coverage and present with productive cough as an initial manifestation, and these factors should be included in the criteria to increase sensitivity of diagnosis among suspected cases of COVID-19.

## Supporting information

**S1 Table. Raw data of all patients under investigation.**
(XLSX)

## Acknowledgments

We thank all Infection Prevention and Control nurses and all staffs at acute respiratory infection clinic at the Faculty of Medicine Ramathibodi Hospital, Mahidol University, Bangkok, Thailand.

## Author Contributions

**Conceptualization:** Jackrapong Bruminhent, Sasisopin Kiertiburanakul.

**Data curation:** Jackrapong Bruminhent, Nattanon Ruangsubvilai, Jeff Nabhindhakara.

**Formal analysis:** Jackrapong Bruminhent, Atiporn Ingsathit, Sasisopin Kiertiburanakul.

**Methodology:** Jackrapong Bruminhent, Sasisopin Kiertiburanakul.

**Validation:** Jackrapong Bruminhent, Atiporn Ingsathit.

**Writing – original draft:** Jackrapong Bruminhent, Nattanon Ruangsubvilai, Jeff Nabhindhakara, Sasisopin Kiertiburanakul.

**Writing – review & editing:** Jackrapong Bruminhent, Sasisopin Kiertiburanakul.

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
