## [Decision Letter · Decision Letter 0]

30 Jul 2020

PONE-D-20-14655

Clinical Characteristics and Risk Factors for Coronavirus Disease 2019 (COVID-19) among Patients under Investigation in Thailand

PLOS ONE

Dear Dr. Kiertiburanakul,

Thank you for submitting your manuscript to PLOS ONE. After careful consideration, we feel that it has merit but does not fully meet PLOS ONE’s publication criteria as it currently stands. Therefore, we invite you to submit a revised version of the manuscript that addresses the points raised during the review process.

ACADEMIC EDITOR: I have received the comments of the reviewers on your manuscript. The specific comments of the reviewers are included below. Please provide point by point response in your revised manuscript.

We look forward to receiving your revised manuscript.

Kind regards,

Muhammad Adrish

Academic Editor

PLOS ONE

Journal Requirements:

2. Please revise your ethics statement on the online submission form to confirm that this study was approved by the Institutional Review Board of the Faculty of Medicine Ramathibodi Hospital, as stated in the methods section of your manuscript.

3. In the ethics statement in the manuscript and in the online submission form, please provide additional information about the patient records used in your retrospective study. Specifically, please ensure that you have discussed whether all data were fully anonymized before you accessed them and/or whether the IRB or ethics committee waived the requirement for informed consent. If patients provided informed written consent to have data from their medical records used in research, please include this information.

4.Thank you for stating the following in your Competing Interests section: 

[No].

Reviewers' comments:

Reviewer's Responses to Questions

**Comments to the Author**

1. Is the manuscript technically sound, and do the data support the conclusions?

Reviewer #1: Yes

Reviewer #2: Yes

2. Has the statistical analysis been performed appropriately and rigorously? 

Reviewer #1: Yes

Reviewer #2: Yes

3. Have the authors made all data underlying the findings in their manuscript fully available?

Reviewer #1: Yes

Reviewer #2: Yes

4. Is the manuscript presented in an intelligible fashion and written in standard English?

Reviewer #1: Yes

Reviewer #2: Yes

5. Review Comments to the Author

Reviewer #1: The article addresses the timely and critical information on the clinical characteristics and risk factors for COVID-19, among the PUIs. The findings of the study add so much of valuable information to medical literature. The article has been written well and data have been analysed appropriately.

However, following are the minor comments to be addressed by the authors:

1. The retrospective review has been done in PUIs covering a period of only 2 weeks. Though the authors have mentioned that this was when there were highest rates of cases reported. It would have added more strength to the analysis had more patients been included.

2. In both the PUIs and the COVID-19 positive population, there is a majority of female population. This is unlike the reports from other countries such as China and India. Can the authors throw more light on this gender difference observed in this study population as against other geographical locations?

3. Patients not being on medical coverage is being found to be an important risk factor owing to the lower socio-economic status and crowded living conditions. That brings up the concern on whether the close contacts of those patients with no medical coverage were traced and tested?

4. In line numbers 105 to 108, there is mention of including cases that did not match the case definition. But there is no mention of the numbers of this category that were included.

5. Also, there is no break up of the numbers of the PUIs being classified as severe and non-severe cases.

6. Line 135 &136 mentions collection of endotracheal aspirates from patients who were intubated. Wondering how many and why were they being intubated even before specimen collection status?

7. What is the fate of the majority of the PUIs tested negative for COVID-19? What could have contributed to the fever, respiratory symptoms among them? Were they tested for other respiratory conditions?

Reviewer #2: This manuscript titiled 'Clinical Characteristics and Risk Factors for Coronavirus Disease 2019 (COVID-19)

among Patients under Investigation in Thailand' is very well written and has useful information in the current situation. Its interesting to see that few of the patients had life threatening, other co-morbidity along with the Covid-19.

I understand that lot of background information is collected from the patients in the hospital. I am not sure if some of it is relevant for the scientific discussion part of this paper. For ex. the religion of the patients. I have hard time figuring out why is it mentioned in the paper if its not discussed anywhere in detail. Same is the case for mentioning percentage of male patient when there is not significant correlation. Thanks.

6. PLOS authors have the option to publish the peer review history of their article (what does this mean?). If published, this will include your full peer review and any attached files.

Reviewer #1: No

Reviewer #2: No

---

## [Author Response · Author response to Decision Letter 0]

13 Aug 2020

Dear the editor and reviewers:

Thank you, the editors and referees, for your review of our manuscript submission and insightful comments. Please see the revised manuscript and response below. 

Sincerely yours, 

Sasisopin Kiertiburanakul, MD, MHS

Reviewer Comments:

Comments to the Author

1. Is the manuscript technically sound, and do the data support the conclusions?

Reviewer #1: Yes

Reviewer #2: Yes

2. Has the statistical analysis been performed appropriately and rigorously? 

Reviewer #1: Yes

Reviewer #2: Yes

3. Have the authors made all data underlying the findings in their manuscript fully available?

Reviewer #1: Yes

Reviewer #2: Yes

4. Is the manuscript presented in an intelligible fashion and written in standard English?

Reviewer #1: Yes

Reviewer #2: Yes

5. Review Comments to the Author

Reviewer #1: The article addresses the timely and critical information on the clinical characteristics and risk factors for COVID-19, among the PUIs. The findings of the study add so much of valuable information to medical literature. The article has been written well and data have been analysed appropriately.

However, following are the minor comments to be addressed by the authors:

1. The retrospective review has been done in PUIs covering a period of only 2 weeks. Though the authors have mentioned that this was when there were highest rates of cases reported. It would have added more strength to the analysis had more patients been included.

Answer: Our study was covered for only 2-week period due to an epidemiological characteristic of COVID-19 in Thailand is relatively brief. Therefore, we tentatively selected this specific period when the majority of PUIs attended healthcare facilities for an investigation when the information regarding some specific risk factors was completely collectable and deem interpretable. We noted this limitation in the discussion section for the authors. See page 16, line 334-338. 

2. In both the PUIs and the COVID-19 positive population, there is a majority of female population. This is unlike the reports from other countries such as China and India. Can the authors throw more light on this gender difference observed in this study population as against other geographical locations?

Answer: We added the following explanation in the discussion section see page 13-14, line 273-279. An unexpected caveat is although the male gender predominance was observed among several cohorts regarding vulnerability to COVID-19, our result instead revealed female gender is more frequently diagnosed. A reason to explain this disparity has been proposed but not entirely clear, however, an outcome seemed indifferent. Furthermore, a greater proportion of female PUIs in our cohort likely from a coincidence or possibly more attention in their health conditions was more prominent among the female population [19, 20].

19) Gebhard C, Regitz-Zagrosek V, Neuhauser HK, Morgan R, Klein SL. Impact of sex and gender on COVID-19 outcomes in Europe. Biol Sex Differ. 2020;11(1):29. Published 2020 May 25. doi:10.1186/s13293-020-00304-9

20) Jin JM, Bai P, He W, et al. Gender Differences in Patients With COVID-19: Focus on Severity and Mortality. Front Public Health. 2020;8:152.

3. Patients not being on medical coverage is being found to be an important risk factor owing to the lower socioeconomic status and crowded living conditions. That brings up the concern on whether the close contacts of those patients with no medical coverage were traced and tested?

Answer: A national policy is also implemented to monitor those who were closely contacted with an index patient (COVID-19 patient) and complimentarily investigated for COVID-19 should new symptoms occur. We added this information in the discussion section. See page 13, line 267-269. 

4. In line numbers 105 to 108, there is mention of including cases that did not match the case definition. But there is no mention of the numbers of this category that were included.

Answer: There were 347 (85.7%) and 58 (14.3%) patients who were fulfilled the criteria and designated as a PUI, respectively. We added this information in the result section. Page 9, line 171-172. 

5. Also, there is no breakup of the numbers of the PUIs being classified as severe and non-severe cases.

Answer: Twenty-six (6.4%) severe PUIs and 379 (93.6%) non-severe PUIs were classified as the aforementioned criteria. We added this information in the result section. Page 9, line 172-173. 

6. Line 135 &136 mentions collection of endotracheal aspirates from patients who were intubated. Wondering how many and why were they being intubated even before specimen collection status?

Answer: Five (1.2%) patients underwent endotracheally intubation on arrival due to acute respiratory failure and therefore endotracheal aspirates were collected accordingly. We added the following sentences “Among 400 (98.8%) patients underwent nasopharyngeal and throat swabs and 5 (1.2%) patients provided endotracheal aspirates for SARS-CoV-2 PCR” We added this information in the result section. Page 9, line 175-176. 

7. What is the fate of the majority of the PUIs tested negative for COVID-19? What could have contributed to the fever, respiratory symptoms among them? Were they tested for other respiratory conditions?

Answer: Among 352 (86.9%) PUIs who did not have a positive result for COVID-19, 40 (11.4%) patients underwent further investigations. There were seven patients in the non-COVID-19 group diagnosed with infections with non-COVID pathogens: influenza virus (n=2), Pneumocystis jirovecii (n=1), Haemophilus influenzae (n=2), Klebsiella pneumoniae (n=1), and Staphylococcus aureus (n=1). We added this information in the result section. Page 11, line 220-224. 

Reviewer #2: This manuscript titled 'Clinical Characteristics and Risk Factors for Coronavirus Disease 2019 (COVID-19) among Patients under Investigation in Thailand' is very well written and has useful information in the current situation. Its interesting to see that few of the patients had life threatening, other co-morbidity along with the COVID-19.

I understand that lot of background information is collected from the patients in the hospital. I am not sure if some of it is relevant for the scientific discussion part of this paper. For ex. the religion of the patients. I have hard time figuring out why is it mentioned in the paper if its not discussed anywhere in detail. Same is the case for mentioning percentage of male patient when there is not significant correlation. Thanks.

Answer: Since there is no correlation of religions and COVID-19, therefore we decided to remove this information. We added the following explanation regarding male sex in the discussion section see page 13-14, line 273-279. An unexpected caveat is although the male gender predominance was observed among several cohorts regarding vulnerability to COVID-19, our result instead revealed female gender is more frequently diagnosed. A reason to explain this disparity has been proposed but not entirely clear, however, an outcome seemed indifferent. Furthermore, a greater proportion of female PUIs in our cohort likely from a coincidence or possibly more attention in their health conditions was more prominent among the female population [19, 20].

19) Gebhard C, Regitz-Zagrosek V, Neuhauser HK, Morgan R, Klein SL. Impact of sex and gender on COVID-19 outcomes in Europe. Biol Sex Differ. 2020;11(1):29. Published 2020 May 25. doi:10.1186/s13293-020-00304-9

20) Jin JM, Bai P, He W, et al. Gender Differences in Patients With COVID-19: Focus on Severity and Mortality. Front Public Health. 2020;8:152.

---

## [Decision Letter · Decision Letter 1]

3 Sep 2020

Clinical Characteristics and Risk Factors for Coronavirus Disease 2019 (COVID-19) among Patients under Investigation in Thailand

PONE-D-20-14655R1

Dear Dr. Kiertiburanakul,

We’re pleased to inform you that your manuscript has been judged scientifically suitable for publication and will be formally accepted for publication once it meets all outstanding technical requirements.

Kind regards,

Muhammad Adrish

Academic Editor

PLOS ONE

Additional Editor Comments (optional):

Reviewers' comments:

Reviewer's Responses to Questions

**Comments to the Author**

1. If the authors have adequately addressed your comments raised in a previous round of review and you feel that this manuscript is now acceptable for publication, you may indicate that here to bypass the “Comments to the Author” section, enter your conflict of interest statement in the “Confidential to Editor” section, and submit your "Accept" recommendation.

Reviewer #1: All comments have been addressed

2. Is the manuscript technically sound, and do the data support the conclusions?

Reviewer #1: Yes

3. Has the statistical analysis been performed appropriately and rigorously? 

Reviewer #1: Yes

4. Have the authors made all data underlying the findings in their manuscript fully available?

Reviewer #1: Yes

5. Is the manuscript presented in an intelligible fashion and written in standard English?

Reviewer #1: Yes

6. Review Comments to the Author

Reviewer #1: The comments raised have been appropriately addressed by the authors in the revised version of manuscript

7. PLOS authors have the option to publish the peer review history of their article (what does this mean?). If published, this will include your full peer review and any attached files.

Reviewer #1: No

---

## [Editor Report · Acceptance letter]

7 Sep 2020

PONE-D-20-14655R1 

Clinical Characteristics and Risk Factors for Coronavirus Disease 2019 (COVID-19) among Patients under Investigation in Thailand 

Dear Dr. Kiertiburanakul:

I'm pleased to inform you that your manuscript has been deemed suitable for publication in PLOS ONE. Congratulations! Your manuscript is now with our production department. 

Kind regards, 

on behalf of

Dr. Muhammad Adrish 

Academic Editor

PLOS ONE